# The Role of a Key Amino Acid Position in Species-Specific Proteinaceous dUTPase Inhibition

**DOI:** 10.3390/biom9060221

**Published:** 2019-06-06

**Authors:** András Benedek, Fanni Temesváry-Kis, Tamjidmaa Khatanbaatar, Ibolya Leveles, Éva Viola Surányi, Judit Eszter Szabó, Lívius Wunderlich, Beáta G. Vértessy

**Affiliations:** 1Budapest University of Technology and Economics, Department of Applied Biotechnology and Food Science, H -1111 Budapest, Szent Gellért tér 4, Hungary; f.temesvary@gmail.com (F.T.-K.); tamjidmaab3@gmail.com (T.K.); leveles.ibolya@ttk.mta.hu (I.L.); suranyi.eva@ttk.mta.hu (É.V.S.); szabo.judit.eszter@ttk.mta.hu (J.E.S.); wunderlich@mail.bme.hu (L.W.); 2Research Centre for Natural Sciences, Hungarian Academy of Sciences, H-1117 Budapest, Magyar tudósok körútja 2, Hungary

**Keywords:** dUTPase, trimer, inhibition, StlSaPIbov1, crystal structure, C-terminal arm

## Abstract

Protein inhibitors of key DNA repair enzymes play an important role in deciphering physiological pathways responsible for genome integrity, and may also be exploited in biomedical research. The staphylococcal repressor StlSaPIbov1 protein was described to be an efficient inhibitor of dUTPase homologues showing a certain degree of species-specificity. In order to provide insight into the inhibition mechanism, in the present study we investigated the interaction of StlSaPIbov1 and *Escherichia coli* dUTPase. Although we observed a strong interaction of these proteins, unexpectedly the *E. coli* dUTPase was not inhibited. Seeking a structural explanation for this phenomenon, we identified a key amino acid position where specific mutations sensitized *E. coli* dUTPase to StlSaPIbov1 inhibition. We solved the three-dimensional (3D) crystal structure of such a mutant in complex with the substrate analogue dUPNPP and surprisingly found that the C-terminal arm of the enzyme, containing the P-loop-like motif was ordered in the structure. This segment was never localized before in any other *E. coli* dUTPase crystal structures. The 3D structure in agreement with solution phase experiments suggested that ordering of the flexible C-terminal segment upon substrate binding is a major factor in defining the sensitivity of *E. coli* dUTPase for StlSaPIbov1 inhibition.

## 1. Introduction

Preservation of genome integrity is of utmost importance for all living organisms. Several complex pathways including numerous enzymes are involved in this duty. Among these, dUTPase and uracil DNA glycosylase enzymes are responsible for keeping DNA uracil-free [1,2,3]. Absence of dUTPase (in differentiated tissues, knock-out models or several microorganisms) [4,5,6,7,8,9] or its inhibition combined with thymidylate synthase-targeted chemotherapies [10,11] is expected to lead to an increased level of uracil in DNA. Under such circumstances, DNA polymerases may readily incorporate uracil into DNA and uracil excision repair is activated. However, if dUTP levels are consistently high, repair cannot be completed efficiently because uracils will again be built into DNA via repair synthesis. Therefore, dUTPase inhibition leads to futile cycles of incomplete repair and may result in increased mutation rate, double strand breaks and thymine-less cell death [12,13,14].

Thymine-less cell death is a clinically significant pathway in tumor chemotherapy where it is usually induced by drugs acting on enzymes being involved in de novo thymidylate biosynthesis (eg. thymidylate synthase or dihydrofolate reductase) [15,16]. Due to the suboptimal response rate of the presently used drugs, development of novel anti-cancer strategies acting on the thymidylate synthesis route is of high significance [17]. The enzyme dUTPase is tightly connected to this pathway, as dUMP is the sole precursor of de novo deoxythymidine triphosphate (dTTP) biosynthesis [14]. Targeting dUTPase by small molecular drugs for cancer treatment has already been reported [13]. This enzyme is also a potential target of future antimicrobiotics against important pathogens such as *Mycobacterium tuberculosis* [7] or *Plasmodium falciparum* [18], as invading pathogen microorganisms require intensive DNA synthesis before cell replication [19].

We recently discovered another potential route for perturbing dUTPase action via a protein inhibitor, Stl_SaPIbov1_ (referred to as Stl in our study) [20]. This protein was originally identified in *Staphylococcus aureus* where its primary function is to repress the transcription of genes being responsible for excision and replication of *S. aureus* pathogenicity islands (SaPi-s). Its protein interaction partners in *S. aureus* are phage dUTPases such as the Φ11 and 80α phage trimeric dUTPases [21]. Upon binding to each other, the primary function of the two proteins is mutually abolished. Namely, in the Φ11 phage dUTPase:Stl complex, Stl is not able to repress transcription of the pathogenicity island protein genes and catalytic activity of the Φ11 phage dUTPase is practically abolished [20].

Previous publications of our research group showed that protein Stl has a potent cross-species inhibitory effect on some other trimeric dUTPase homologues from *Mycobacteria*, *Drosophila* and human sources as well [22,23,24]. Efficiency of Stl-induced inhibition on dUTPase activity varies among the complexes with different trimeric dUTPase homologues, and the exact structural background of these species-specific differences in Stl binding are yet to be investigated in details. The in-depth structural explanation for the alterations in inhibitory capacity would present a major step forward in the future development of species-specific dUTPase inhibitory peptides or peptide-mimicking small molecules [19]. Stl may also be a useful tool for an in vivo study of the effects of the elimination of dUTPase activity. The understanding of the structural background of species-specific inhibitory effect of Stl on different dUTPases would make it possible to predict whether Stl will be applicable as a tool to study a given dUTPase in vivo.

In our previous studies, we reported insights into Stl-inhibition of dUTPases from staphylococcal, human, mycobacterial and *Drosophila* sources [20,22,23,24]. We observed a generally valid inhibition pattern characterized by a strong dUTPase:Stl complex with dissociation constant in the nanomolar range, and provided a detailed elucidation of the molecular mechanism of interaction and inhibition for the Φ11 dUTPase:Stl system [20]. However, a full understanding of the structural biology of the Stl-dUTPase complex is not yet available, since on the one hand crystallization trials of the complex have not yet been successful, and on the other hand the molecular size of the complex prevents high-resolution nuclear magnetic resonance (NMR) studies. For the protein Stl, no experimentally determined three-dimensional (3D) structure has yet been published. In silico homology modelling of the Stl structure accompanied by synchrotron radiation circular dichroism studies suggested a protein fold mostly containing alpha-helices [25]. For the protein dUTPase, there are numerous high-resolution dUTPase crystal structures deposited in the Protein Data Bank (PDB) (reviewed eg. in [2]). These structures all show the well-conserved dUTPase jelly-roll fold; however, there are some structural variations. Interestingly, in the *Escherichia coli* dUTPase structures deposited so far very few residues from the C-terminal arm of the enzyme can be localized, probably due to high conformational flexibility in this region [26]. The dUTPase C-terminal arm was previously shown to play some, although not essential role in Stl binding [20,27]. In the present work, we extended our Stl inhibition studies to *E. coli* dUTPase, since from this aspect it seems to be different from the Φ11 bacteriophage, human or mycobacterial enzymes for which Stl inhibition was observed [28,29,30].

Here we show that *E. coli* dUTPase and protein Stl form a strong protein complex associated with nanomolar dissociation constant. Unexpectedly, we also found that Stl inhibition of *E. coli* dUTPase in steady-state activity measurements could not be observed, therefore this enzyme presents a potentially useful model for further understanding of the governing factors leading toward dUTPase inhibition by Stl. Based on previous structural data we initiated a series of rationally designed mutations to reveal the key components involved in inhibition. Two such mutations were successfully identified (mutation of the 93rd glutamine to either histidine or arginine, abbreviated as EcDUT^Q93H^ and EcDUT^Q93R^, respectively). We found that both of these mutant *E. coli* dUTPases were significantly inhibited by Stl. We have also crystallized the EcDUT^Q93H^ mutant in complex with a non-hydrolysable substrate analogue, α,β-imido-dUTP (dUPNPP) and solved the 3D structure of this complex. The detailed structural information combined with solution experiments provided insights into the potential role of C-terminal arm flexibility and Stl-induced inhibition.

## 2. Materials and Methods 

### 2.1. Site-Directed Mutagenesis

Site-directed mutagenesis was carried out based on either the original QuikChange mutagenesis protocol (Agilent Technologies, Santa Clara, CA, USA) or on a modified version using only partially overlapping primers [31]. The sequence of the primers used in our polymerase chain reactions (PCRs) is listed in Table 1.

### 2.2. Molecular Cloning, Protein Expression and Purification

The gene encoding protein Stl was expressed from a pGEX-4T-1 vector (GE Healthcare, Chicago, Illinois, USA) as a glutathione S-transferase (GST)-fused construct. The GST-tag was cleaved from the protein via overnight thrombin digestion during its purification process [24]. The *E. coli* dUTPase gene was inserted into a pET-15b vector (Merck KGaA, Darmstadt, Germany) between the BamHI and NdeI cleavage sites to enable its expression with an N-terminal His-tag which was used in further purification steps [32]. Both protein Stl and our *E. coli* dUTPase constructs were expressed in *E. coli* BL21 (DE3) Rosetta cells (Novagen) and they were purified according to our previously used protocol [24].

### 2.3. Size Exclusion Chromatography

For this measurement, an AKTA FPLC purification system with a Superose 12 10/300 GL column (GE Healthcare) was used. EcDUT, protein Stl and their 1:1 monomeric molar ratio mixture were injected on the column in dUTPase buffer (20 mM HEPES (4-(2-hydroxyethyl)-1-piperazineethanesulfonic acid), 300 mM NaCl, 5 mM MgCl2, 10mM β-mercaptoethanol, pH = 7.5). Peak elution volumes of the three separate injections were compared and plotted on the same graph.

### 2.4. Differential Scanning Fluorimetry (Thermofluor)

Samples were heated from 25 to 85 °C in a BioRad CFX96 Touch instrument (Hercules, CA, USA) using three parallels of each measurement in 25 µL volumes. EcDUT and protein Stl were used in 1:1 molar ratio in their mixture and single protein concentrations were set to 40 µM, corresponding to monomeric protein subunits. Sypro Orange dye (ThermoFisher Scientific, Waltham, MA, USA) was added to the samples in 1000-fold dilution to follow protein unfolding. Melting points were determined as the extremum values corresponding to the first negative derivate of the melting curve (cf. [24]).

### 2.5. Multiple Sequence Alignment Based Mutational Screen

The dUTPase protein sequences were obtained from the Protein Data Bank and the UniProt database [33,34]. Sequence alignments were carried out using the Clustal Omega server [35]. Conserved dUTPase sequence motifs were identified based on earlier studies on dUTPases [2,26]. Recent hydrogen deuterium exchange-mass spectrometry (HDX-MS) measurements on the interaction surface of human dUTPase and protein Stl were also taken into consideration in the sequence alignment [23].

### 2.6. Measurement of Steady-State Enzyme Activity and Inhibition

The by-product of dUTP hydrolysis is a proton released which causes a change in the pH of the reaction mixture. Adding phenol-red indicator to the reaction buffer (1 mM HEPES, 150 mM KCl, 5 mM MgCl_2_, 40 µM phenol red, pH = 7.5) enables quantification of dUTP hydrolysis. Change in absorbance upon dUTP addition was followed at 559 nm at 20 °C. Quasi steady-state velocity (v_0_) was determined by fitting a curve to the linear phase of the progress line. Stl inhibition measurements were carried out after 5 min pre-incubation of the Stl-dUTPase mixture at 20 °C. The concentration of dUTPase was kept constant at 50 nM, while Stl concentration varied between 0 and 400 nM. Always 30 µM dUTP was used to initiate the enzymatic reaction after pre-incubation of the proteins. Three parallels were measured in each cases [24]. For plotting relative k_cat_ values, propagation of standard errors was taken into consideration according to the following formula:(1)$${\mathrm{SD}}_{F}^{2}=\frac{k_{\mathrm{cat},\;400}^{2}}{k_{\mathrm{cat},\;0}^{2}}*\left(\frac{{\mathrm{SD}}_{400}^{2}}{k_{\mathrm{cat},\;400}^{2}}+\frac{{\mathrm{SD}}_{0}^{2}}{k_{\mathrm{cat},\;0}^{2}}\right)$$
where SD_F_ is the propagated standard deviation, k_cat,0 and_ k_cat,400_ is the catalytic rate constant in absence and presence of 400 nM Stl, and SD_0_ and SD_400_ are their respective standard deviations.

A quadratic equation was fitted on the Stl inhibition curves of the EcDUT^Q93H^ and EcDUT^Q93R^ mutants according to the following formula:(2)$$y=s+\;\frac{A[\left(c+x+K\right)-\sqrt{{\left(c+x+K\right)}^{2}-4\mathrm{cx}]}}{2c}$$
where y is the (relative) steady-state enzyme activity, x is protein Stl’s concentration, s is the (relative) steady-state enzyme activity without Stl addition, A is the total decrease in (relative) steady-state activity, c is dUTPase concentration (kept constant) and K is the K_i_ (IC_50_) value which is the output parameter obtained from curve fitting. 

### 2.7. Isothermal Titration Calorimetry

Isothermal tritation calorimetry (ITC) experiments were carried out at 20 °C on a Microcal ITC200 instrument (Malvern Instruments, Malvern, UK). The proteins were dialysed against a buffer pH = 7.5 comprising 20 mM HEPES, 300 mM NaCl and 1 mM TCEP. We used 22–57 μM Stl in the cell and 330–550 μM enzyme (EcDUT, EcDUT^Q93H^, EcDUT^Q93R^) in the syringe. Both protein concentrations correspond to monomeric subunits. The titrations were performed with the injection syringe rotating at 750 rpm (revolutions per minute) and included a series of 20 injections spaced 180 s apart from each other, with injection volumes of 0.5 μL for the first titration and 2 μL for the subsequent 19 titrations. The data were analyzed using Microcal Origin software following the directions of the manufacturer (Malvern Panalytical Ltd, Malvern, UK). The one set of independent sites binding model was applied to data for determination of thermodynamic parameters: dissociation constant (Kd), stoichiometry (N), enthalpy (ΔH) and entropy (ΔS). The mean and standard deviation (SD) of the parameters were calculated from three independent experiments.

### 2.8. Protein Crystallization and Structural Refinement

EcDUTQ93H was gel filtrated in 20 mM HEPES, 100 mM NaCl, 5 mM MgCl2, pH = 7.5 (gel filtration buffer) and then immediately concentrated to 43 mg/mL and mixed with 5 mM dUPNPP before crystallization. The enzyme–dUPNPP complex was incubated on ice for at least 30 min before being mixed with the reservoir solution (0.1 M TRIS, 18–33.75 % polyethylene glycol 3350, 400 mM NaAc, pH = 7.5) in 2:1 or 1:1 protein-reservoir ratios. Crystals were grown with hanging drop vapor diffusion method at 295 K. Data were collected at the European Synchrotron Radiation Facility (ESRF, Grenoble, France) at Beamline ID30A-3. For data collection, a cryoprotectant containing 12% glycerol was used. For data processing and scaling, the XDS program package was used [36].

The crystal structure was solved by molecular replacement using a wild-type *E. coli* dUTPase crystal structure in complex with dUPNPP (PDB-ID:1RN8) as a template [26]. For structural refinement, the program packages PHENIX and CCP4 were used [37,38,39,40]. Data collection and refinement statistics are summarized in Table 2. Three monomers are present in the asymmetric unit. Coordinates and crystal factor data are deposited in Protein Data Bank with the identification code 6HDE. Figures were created using PyMOL [41].

### 2.9. Tryptophan Fluorimetry

Samples of 3 µM dUTPase in 20 mM HEPES, 300 mM NaCl, 5 mM MgCl2, pH = 7.5 were prepared. Protein Stl or dUPNPP were added in 4.5 µM or 100 µM final concentration, respectively. Tryptophan fluorescence spectra were recorded at 293 K on a Jobin Yvon Spex FluoroMax-3 spectrofluorometer (Horiba France SAS, Palaiseau, France) between 300 nm and 400 nm, using 295 nm excitation wavelength. Excitation and emission slits were set to 1 nm and 5 nm, respectively. Fluorescence spectrum of the assay buffer was subtracted from all protein spectra to eliminate additional fluorescence or inner filter effects.

### 2.10. Acrylamide Quenching and Statistical Analysis

Acrylamide quenching was measured on a BioTek Synegry MX plate reader (BioTek Instruments, Inc. Winooski, Vermont, USA) on 96-well plates in 25 µL final volumes using 4 µM dUTPase concentration. Both the EcDUT^F145W^ and EcDUT^Q93H,F145W^ enzymes were measured in order to compare their C-terminal arm movements. A titration of the enzyme, enzyme-dUPNPP, enzyme-Stl or *N*-acetyl-L-tryptophanamide (NATA) solutions was carried out with high purity acrylamide (Sigma-Aldrich, cat. no.: A9099, subsidiary of Merck KGaA, St. Louis, Missouri, USA) in 0–0.40 µM concentration range on three parallel plates ensuring three independent titration curves of the compared protein samples. Samples were excited at 295 nm and their emission was measured at 338 nm. Protein Stl or dUPNPP was added to the respective samples in 6 µM or 1.5 mM final concentration. Fluorescence of the acrylamide solution itself was subtracted from all titration curves. A modified Stern-Volmer equation was fitted to the titration curves (Equation (3)).
(3)$$\frac{F_{0}}{F}=1+K_{\mathrm{SV}}*\left[Q\right]*e^{V\left[Q\right]}$$

In the equation F_0_ is the unquenched and F is the quenched tryptophan fluorescence, Q is the quencher (acrylamide), K_sv_ is the dynamic (bimolecular) quenching constant and V is the static (sphere of action) component of quenching [42,43,44].

Analysis of variance (ANOVA) was carried out on the obtained K_sv_ data by using Statistica 13 software. The ANOVA was followed by two planned comparisons (*t* tests) between the apo and dUPNPP-bound states of the EcDUT^F145W^ and EcDUT^Q93H,F145W^ enzymes. Significance level was set at 5% using two-sided *p* values.

## 3. Results and discussion

### 3.1. Stl Binds to Escherichia coli dUTPase but Does Not Show Inhibitory Potential in Steady-State Kinetic Experiments

To decide whether Stl may bind to *E. coli* dUTPase (EcDUT) we have performed a size-exclusion chromatographic experiment. Figure 1A shows that a mixture of Stl and EcDUT elutes from the size-exclusion chromatographic column in a symmetric single peak, clearly separate from the elution peaks of Stl or EcDUT alone. This peak is associated with a lower elution volume (i.e. higher molecular weight) arguing for the formation of a protein-protein complex. Complexation of these two proteins was also investigated using differential scanning fluorimetry for both the components (Stl and EcDUT) and their mixture (complex). Differential scanning fluorimetry (also known as thermofluor) is a straightforward method to follow the thermal denaturation profile of proteins on their own or in complexes and to determine their melting temperature (T_m_) [45]. As shown on Figure 1B, EcDUT and Stl show different T_m_ values (70.5 °C and 52.0 °C, respectively), whereas the T_m_ observed in their complex is 57.0 °C.

Having established the potential of complex formation between EcDUT and Stl, we next investigated if the enzymatic activity of dUTPase is inhibited in the protein complex. For several dUTPases from different sources, Stl was reported to show an inhibitory effect on dUTPase activity [20,22,23,24]. It was therefore unexpected to observe that addition of Stl to the reaction mixture during steady-state enzyme activity measurements did not result in an inhibitory pattern (see data on Figure 2A,B).

### 3.2. Searching for Key Residues in Inhibition and Identification of Such a Position

It was of immediate interest to determine the structural background for this unexpected lack of inhibition of EcDUT by Stl. Towards this end we started with investigation of differences in the primary structure of EcDUT as compared to all the other dUTPases where Stl was shown to be a protein inhibitor (Figure 2C). Our rational was to locate residues that are conserved in all the other dUTPases which are inhibited by Stl, but not in EcDUT. We also included residues suggested to be relevant in this aspect from comparisons between the complexes formed by Stl and different phage dUTPases (80α and Φ11 phage dUTPases) [27]. These residues are indicated on a green background on Figure 2C. We then performed rationally designed mutations where the residue in the EcDUT protein was exchanged for the residue observed in another dUTPase inhibitable by Stl. Results are shown on Figure 2A and are also summarized on the inset table of Figure 2B.

Several of these mutations (EcDUT^E114D^, EcDUT^E114D, R115K^, EcDUT^H147S^) did not lead to any observable inhibition by Stl. However, we could successfully locate one specific position (EcDUT^Q93^) where mutation of the glutamine side-chain either into histidine or arginine led to strong inhibition of dUTPase activity by Stl (Figure 2A,B). Histidine at this position was found in the Φ11 dUTPase, whereas in *M. tuberculosis, Drosophila melanogaster* and *Homo sapiens* dUTPases, arginine is present at this site. Interestingly, as shown in recent HDX-MS experiments [23], this side-chain position is located within a peptide segment involved in formation of the interaction surface of the human dUTPase-Stl protein complex.

The successful modification of the *E. coli* enzyme structure clearly resulted in a character being inhibitable by Stl, arguing for the key importance of the mutated residue position in the inhibitory mechanism. The two resulting mutant constructs (EcDUT^Q93H^ and EcDUT^Q93R^) were both clearly inhibited by Stl and the inhibition resulted in a decrease of the steady-state velocity by 58% and 47%, respectively.

We also wished to quantify the strength of interaction of the complexes of wild-type EcDUT and the mutant EcDUT constructs with Stl. Figure 3 shows the results of isothermal titration microcalorimetric experiments. Based on these data we observe that mutation of the 93rd glutamine position into either arginine or histidine increases the strength of interaction by approximately two-three fold. Although the dG values for the complexes involving the wild-type and the mutant dUTPases are very similar, it is important to note that dH values are significantly higher for the EcDUT^Q93H^:Stl and EcDUT^Q93R^:Stl complexes. This finding indicates the creation of additional favorable enthalpic interactions for these complexes (potential H-bonds, polar or Van der Waals interactions).

### 3.3. Understanding the Structural Background of Inhibitablity

It was of clear interest to understand the exact changes in the enzyme structure caused by the point mutations at the 93rd amino acid position. Therefore, we crystallized the EcDUT^Q93H^ dUTPase mutant in the presence of the non-hydrolysable substrate analogue dUPNPP and compared its structure to the already deposited wild-type *E. coli* dUTPase structures (Figure 4). The 3D structure of the EcDUT^Q93H^ dUTPase mutant (PDB ID 6HDE) is shown on Figure 4B–G.

A novelty of our structure is that the whole C-terminal arm of the enzyme is visible in one of the subunits, whereas major parts of the segment are also visible in the other two subunits. This part of the enzyme has central role in catalysis, however it was so far missing from all existing *E. coli* dUTPase structures, even if they were crystallized in the presence of a substrate analogue. Our crystal structure provides a strong implication for existing interactions among the histidine point mutation at the 93rd position and the C-terminal arm of the enzyme, serving with a straightforward explanation for increased visibility of the C-terminal arm section.

We hypothesized that the increased sensitivity of the EcDUT^Q93H^ and EcDUT^Q93R^ mutants for inhibition by protein Stl and their stronger Stl binding ability compared to the wild-type *E. coli* dUTPase is in connection with the restricted conformational freedom of the C-terminal arm of the enzyme.

To find further explanation for the fact that the EcDUT^Q93H^ and EcDUT^Q93R^ point mutants can be readily inhibited by Stl, and to test our arm-flexibility hypothesis, we were intended to establish a technique to follow C-terminal arm movements upon substrate or Stl binding. Towards this end, we introduced a tryptophan point mutation into the C-terminal arms of the wild-type, EcDUT^Q93H^ and EcDUT^Q93R^ mutant dUTPases, obtaining three additional enzyme variants, namely EcDUT^F145W^, EcDUT^Q93H, F145W^ and EcDUT^Q93R, F145W^.

The engineered Trp residue indeed allowed us to follow substrate analogue and Stl binding of the three arm-tryptophan mutant enzyme variants (Figure 5 and cf. Figure 2B) [28]. In comparison with earlier published studies we therefore confirm here that binding of the cognate ligand to the dUTPase active site is readily transmitted to decreased fluorescence intensity of the tryptophan fluorophore within the C-terminal arm [28,30,46] (cf. Figure 2A).

To enable a more direct comparison of the arm movement behavior of our tryptophan sensor containing mutants, acrylamide quenching experiments were planned. The K_sv_ constant values obtained by this method revealed that the quasi wild-type EcDUT^F145W^ and EcDUT^Q93H,F145W^ mutant enzymes are remarkably different in their arm movements upon dUPNPP substrate analogue binding (Figure 6A,B). Data from the quenching experiments clearly argue that solvent accessibility of the tryptophan residue in the EcDUT^Q93H,F145W^ dUTPase – dUPNPP complex is decreased as compared to the complex formed with EcDUT^F145W^. Difference in solvent accessibility (K_sv_ values) between the apo and dUPNPP-bound state of the EcDUT^Q93H,F145W^ mutant enzyme was proved to be statistically significant (*p* = 0.0073, cf. Figure 6A), while in the case of the EcDUT^F145W^ enzyme no significant difference was observed (*p* = 0.5957). We therefore conclude that the quenching experiments in the solution state provide re-enforcement for the restricted conformational flexibility of the C-terminus observed in the crystal structure.

## 4. Conclusions

In conclusion, we have demonstrated that specific side-chain mutations dramatically alter the character of interaction between Stl and *E. coli* dUTPase. Namely, we have found that changing the wild-type glutamine into either arginine or histidine at the 93rd position leads to an enzyme variant which is strongly inhibited by Stl. We also found that these mutations alter the extent of orderliness in the C-terminal segment which is achieved by substrate binding to the *E. coli* enzyme. These data suggest a link between C-terminal arm flexibility and inhibition characteristics of *E. coli* dUTPase. 

## Figures and Tables

**Figure 1 biomolecules-09-00221-f001:**
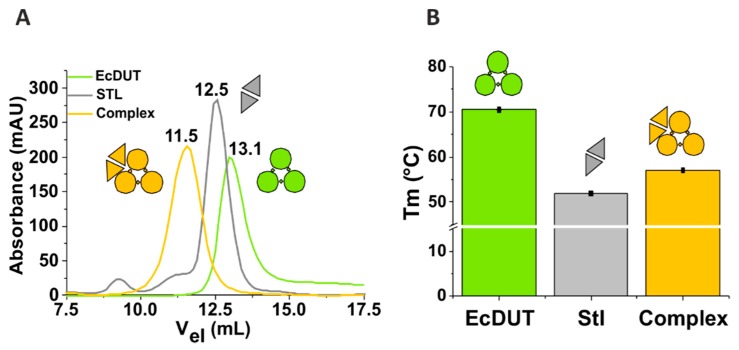
Complex formation between EcDUT and Stl. Data and symbols for EcDUT, Stl and their mixtures are shown in green, grey, and yellow colors, respectively. Pictograms show the oligomeric assembly characteristic for each species. (**A**) Size-exclusion chromatography. Peak elution volumes are marked with numbers (data in mL) above the elution peaks. (**B**) Differential scanning fluorimetry. Melting points are plotted as mean values with standard deviations (less than 0.5 °C for all measurement replicates).

**Figure 2 biomolecules-09-00221-f002:**
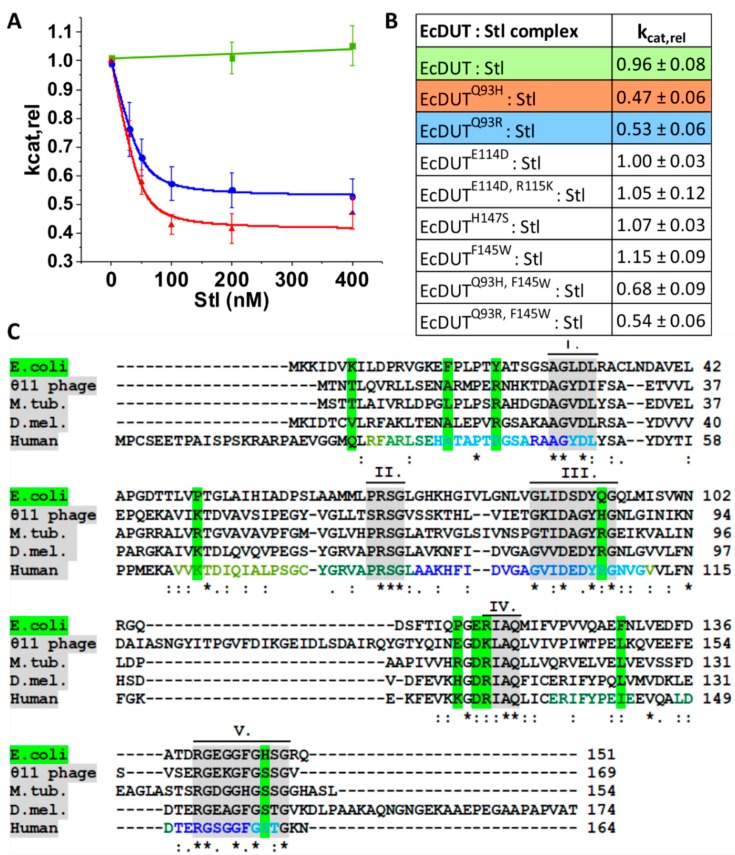
Identification of key residues in dUTPase inhibition upon Stl binding. (**A**) Stl inhibits the EcDUT^Q93H^ and EcDUT^Q93R^ point mutants but not the wild-type EcDUT. Steady-state activities of EcDUT (green), EcDUT^Q93R^ (blue) and EcDUT^Q93H^ (red) against Stl concentration are plotted on the graph. A quadratic equation was fitted on the inhibitable enzymes, from which K_i_ (IC_50_) values were calculated. K_i_ was measured to be 4.83 ± 3.46 nM for EcDUT^Q93H^ and 5.90 ± 0.73 for EcDUT^Q93R^, respectively. (**B**) Relative k_cat_ values of EcDUT constructs upon 400 nM Stl addition, referred to the activity measured without Stl addition. For standard deviation values, propagation of uncertainty upon normalization was taken into consideration (cf. Materials and Methods). (**C**) Multiple sequence alignment comparing the non-inhibitable (*Escherichia coli*) and inhibitable (Φ11 phage, *Mycobacterium tuberculosis*, *Drosophila melanogaster*, human) dUTPases. Conserved dUTPase sequence motifs contributing for active site architecture are numbered (I–V) and shown on grey background. Altered side-chain characteristics of EcDUT compared to the inhibitable dUTPases are shown on green background. Interaction surface of human dUTPase with protein Stl determined by recent hydrogen/deuterium exchange-mass spectrometry (HDX-MS) measurements is highlighted on the human dUTPase sequence with the same color code as it was published in ref. [23].

**Figure 3 biomolecules-09-00221-f003:**
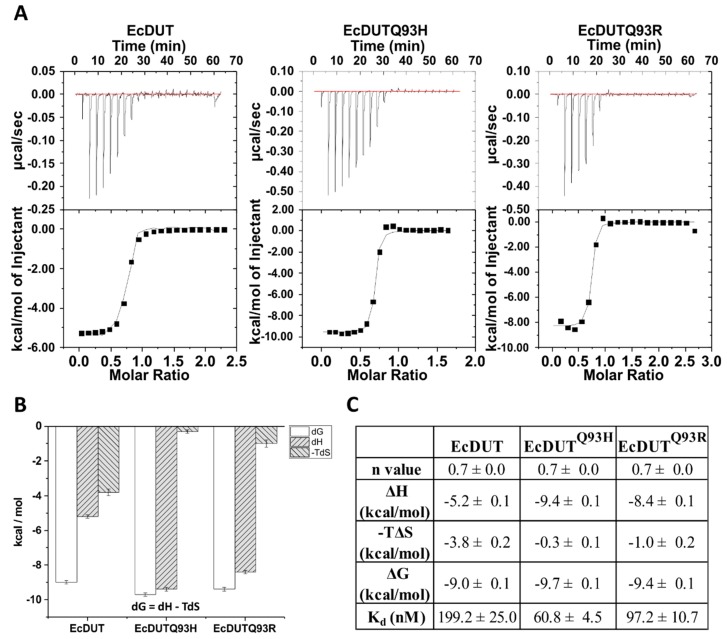
Thermodynamic characterization of EcDUT:Stl complex formation. (**A**) Titration of EcDUT (left), EcDUT^Q93H^ (middle) and EcDUT^Q93R^ (right) with protein Stl. (**B**) and (**C**) Thermodynamic data of complex formation.

**Figure 4 biomolecules-09-00221-f004:**
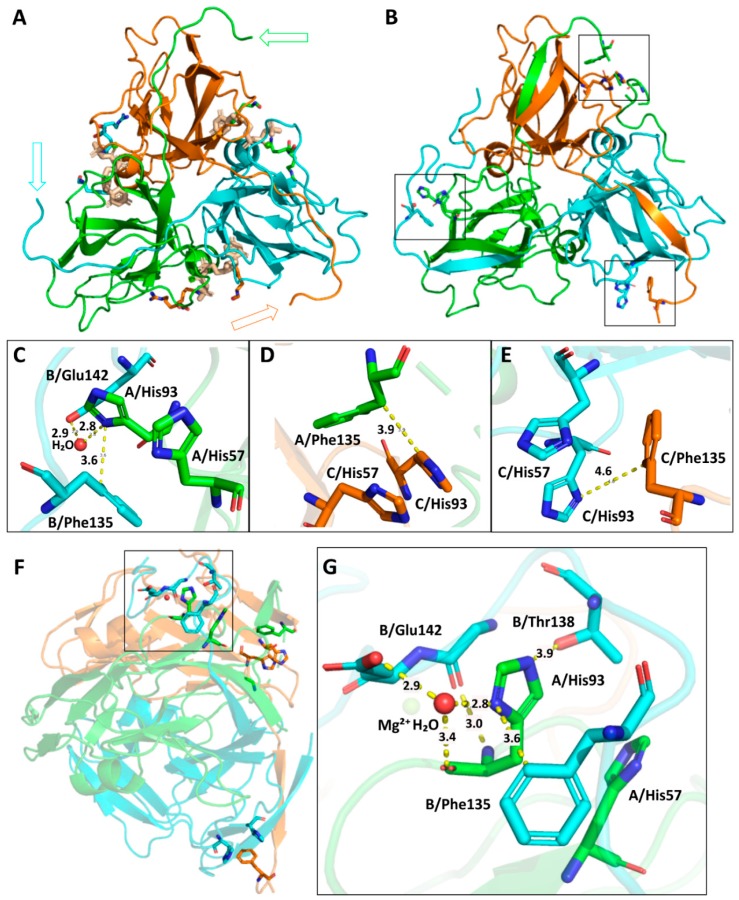
Structural insights into mutation-induced conformational changes. Subunits of the enzyme are color coded (orange, green, blue). The three dUPNPP molecules at the active sites are highlighted with wheat-colored sticks. (**A**) Wild-type EcDUT crystal structure (PDB-ID:1RN8) [26] that was used as a template in our molecular replacement. Missing parts of the C-terminal arms are indicated with empty arrows. Only F145, R150 and Q151 are visible out of the last 15 C-terminal residues of the 1RN8 model, visualized by line representation. (**B**) The presently determined EcDUT^Q93H^ crystal structure (PDB ID:6HDE). Positions of the mutated H93 residues are marked with boxes. (**C**), (**D**), (**E**) The point-mutant H93 side-chain can establish aromatic contacts with the C-terminal arm of the neighboring subunit. (**F**) The EcDUT^Q93H^ crystal structure from side-view. (**G**) A water-linked hydrogen-bonding contact between the mutated H93 residue and the Q142 side-chain of the C-terminal arm.

**Figure 5 biomolecules-09-00221-f005:**
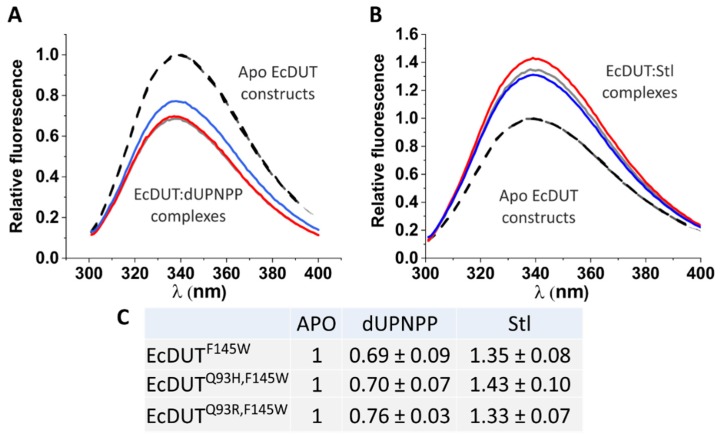
The active site tryptophan sensor reports on ligand binding to EcDUT. (**A**) Binding of the substrate analogue dUPNPP. (**B**) Binding of protein Stl. (**C**) Comparison of peak relative fluorescence values. Black dashed lines stand for the EcDUT^F145W^, EcDUT^Q93H,F145W^ and EcDUT^Q93R,F145W^ apoenzyme constructs. Ligand bound EcDUT^F145W^, EcDUT^Q93H,F145W^ and EcDUT^Q93R,F145W^ are represented by grey, red and blue straight lines, respectively.

**Figure 6 biomolecules-09-00221-f006:**
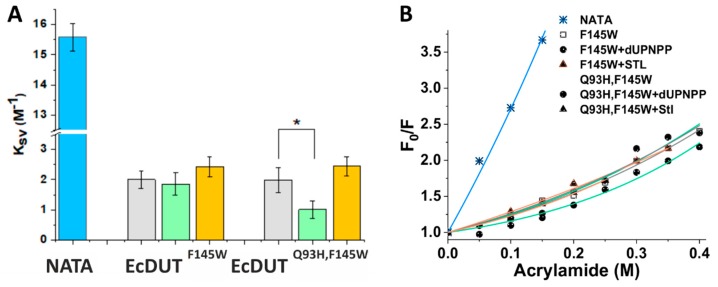
Acrylamide quenching reveals altered solvent accessibility upon mutation in EcDUT. (**A**) Comparison of quenching constants (K_sv_ values). Please note that lower K_sv_ values argue for less solvent exposed C-terminal arms. Blue color refers to NATA solution (reference for the quenching constant of free tryptophan residues in the solution). The apo-state fluorescence of EcDUT^F145W^ and EcDUT^Q93H,F145W^ (green) was compared to their dUPNPP-bound (grey) and Stl-bound (orange) state. A star above the respective bars indicates that K_sv_ difference between the apo- and dUPNPP-bound state is statistically significant for EcDUT^Q93H,F145W^ but not for the EcDUT^F145W^ enzyme construct. (**B**) F_0_/F values as a function of acrylamide concentration.

**Table 1 biomolecules-09-00221-t001:** List of primers used for site-directed mutagenesis. Mismatched nucleotides are marked with red letters.

**Q93H _FW**	5’ GATCGATTCTGACTATCATGGCCAGTTGATGATTTCC 3’
**Q93H_REV**	5’ GGAAATCATCAACTGGCCATGATAGTCAGAATCGATC 3’
**Q93R_fw**	5’ GACTATCGGGGCCAGTTGATGATTTCCGTGTGG 3’
**Q93R_rev**	5’ CTGGCCCCGATAGTCAGAATCGATCAATCCTACCAGG 3’
**E114D_fw**	5’ CATTCAACCTGGCGATCGCATCGCCCAG 3’
**E114D_rev**	5’ CTGGGCGATGCGATCGCCAGGTTGAATG 3’
**R115K_fw**	5’ CCTGGCGAAAAAATCGCCCAGATGATTTTTGTTCCGGTAGTACAGGCTGAATTTAATCTGGTGG 3’
**R115K_rev**	5’ CTGGGCGATTTTTTCGCCAGGTTGAATGGTGAAGCTGTCCTGACCACGG 3’
**ER114-5DK_fw**	5’ CAACCTGGCGATAAAATCGCCCAGATGATTTTTGTTCCGGTAGTACAGGCTGAATTTAATCTGGTGG 3’
**ER114-5DK_rev**	5’ CATCTGGGCGATTTTATCGCCAGGTTGAATGGTGAAGCTGTCCTGACCACGG 3’
**F145W_fw**	5’ GAAGGCGGCTGGGGTCACTCTGGTCGTCAGTAACACATACGGATCCGGC 3’
**F145W_rev**	5’ AGAGTGACCCCAGCCGCCTTCACCGCGGTCGGTGGCGTC 3’
**H147S_fw**	5’ CCGCGGTGAAGGCGGCTTTGGTAGCTCTGGTCGTCAGTAACAC 3’
**H147S_rev**	5’ GTGTTACTGACGACCAGAGCTACCAAAGCCGCCTTCACCGCGG 3’

**Table 2 biomolecules-09-00221-t002:** Data collection and refinement statistics for the EcDUT^Q93H^ structure (PDB-ID:6HDE).

Data Collection	Parameters
Space group	P 21 21 21
Unit-cell parameters (A°)	a = 63.20, b = 66.50, c = 95.30
Unit-cell parameters (angles)	α = β = γ = 90°
Resolution range (A°)	45.811-1.8
Total No. of reflections	113402
No. of unique reflections	36491
Completeness (%)	99.2
<I/σ(I)>	1.64 (at 1.82 Å)
Rmeas	0.043
**Refinement**	
No. of dUTPase subunits in asymmetric unit	3
No. of protein atoms	3257
No. of ligand atoms	84
No. of waters	162
No. of Mg^2+^ ions	3
Rcryst/Rfree‡	0.1806/0.2209
Average B factors (Å^2^) (all atoms)	33.0
Wilson B factor (Å^2^)	27.4
Protein atoms	3257
Ligand atoms	84
Water	162
Mg^2+^ ions	3
R.m.s. deviations from ideal values	
Bond lengths (A°)	0.007
Bond angles (°)	1.01
**Ramachandan plot analysis, residues in (%)**	
Favoured region	96.97
Allowed region	3.03
Disallowed region	0.00

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
