# Peer review of "The Role of a Key Amino Acid Position in Species-Specific Proteinaceous dUTPase Inhibition"

_biomolecules, 2019, doi:10.3390/biom9060221_

Round 1

Reviewer 1 Report

The authors have investigated the role of position of Key Amino Acids in protein- protein interaction of StlSaPIbov1 and dUTPase, two important proteins in DNA repair system. They have properly used different structural biology and biophysical methods in this protein structure-function investigation.

The introduction covers both background and previous published works. Selection of employed methods in their research is very proper and fit for their approach in answering raised questions in this particular research. Based on explanations in the method section, it seems they have performed each experiment properly and also, they have included all of required details in their manuscript.

The results have been depicted and represented very well and professionally. The authors discussed clearly their results. The English language of paper is good and fluent.  I believe this research is a very standard and clean-cut protein structure-function study that correctly utilized relevant methods accompanied by proper statistical data analysis.  

Reviewer 2 Report

In this manuscript, the authors identify a E.coli dUTPase that associates with the known dUTPase inhibitor StlSaPIbov1, but doesnot inhibit the dUTPase. The authors identified specific amino acid positions in E.coli dUTPase that sensitized the enzyme to Stl.inhibition. This manuscript is well written and the conclusions are supported by the reults.

Minor comments:

 Line 17 says a strong interaction between the proteins, can the authors verify this on native gel electrophoresis ? Becasue the lack of inhibition could be due to a transient, weak interaction and as later shown in Figure 3, specific amino acid mutations strengthen the interaction.Can the Stl complexes with Wt versus mutant identified on native gel electrophoreis ?